# Is the Calcium Score Useful for Rheumatoid Arthritis Patients at Low or Intermediate Cardiovascular Risk?

**DOI:** 10.3390/jcm11164841

**Published:** 2022-08-18

**Authors:** Claire Jesson, Yohann Bohbot, Simon Soudet, Cedric Renard, Jean-Marc Sobhy Danial, Laetitia Diep, Marie Doussière, Christophe Tribouilloy, Vincent Goëb

**Affiliations:** 1Department of Rheumatology, Amiens University Hospital, Université de Picardie Jules Verne (UPJV), 80000 Amiens, France; 2Department of Cardiology, Amiens University Hospital, Université de Picardie Jules Verne (UPJV), 80000 Amiens, France; 3Department of Vascular Medicine, Amiens University Hospital, Université de Picardie Jules Verne (UPJV), 80000 Amiens, France; 4Department of Radiology, Amiens University Hospital, Université de Picardie Jules Verne (UPJV), 80000 Amiens, France

**Keywords:** rheumatoid arthritis, cardiovascular risk, atherosclerosis, coronary artery calcification score, primary prevention

## Abstract

Cardiovascular disease, particularly myocardial infarction, is the leading cause of death of rheumatoid arthritis (RA) patients. The usefulness of the coronary artery calcification score (CACS), determined using cardiac computed-tomography (CT)-scan images, was assessed as a part of a cardiovascular work-up of RA patients at low or intermediate cardiovascular disease risk. This descriptive, cross-sectional, single-center study was conducted on patients with stable RA or that which is in remission. Each patient’s work-up included a collection of cardiovascular risk factors, laboratory analyses, an electrocardiogram, a supra-aortic trunks (SATs) echo-Doppler test and a cardiac CT scan. The primary endpoint was to determine the frequency of patients with a CACS > 100, indicating notable atherosclerosis. Fifty patients were analyzed: mean ± standard deviation age was 53.7 ± 7.5 years, 82% women. The CACS exceeded 100 in 12 (24%) patients (11 were at intermediate risk) and 2 of them underwent angioplasty for silent myocardial ischemia. Cardiovascular risk was reclassified from intermediate to high for 5 patients. Age according to sex and smoking status were significantly associated with that increase; no association was found with RA characteristics or treatments.

## 1. Introduction

Rheumatoid arthritis (RA) is the most common chronic inflammatory rheumatic disease [1]. Other than its rheumatological consequences, patients are at a higher risk of developing cardiovascular disease. This higher prevalence is usually explained by RA activity or its treatment [2]. Myocardial infarction (MI) is the main cause of RA patient mortality [3]. Assessment of RA patients’ cardiovascular risk is an integral part of the management of this chronic disease. It is currently recommended that researchers collect conventional cardiovascular risk factors and perform a lipid profile and a SATs echo-Doppler test [4]. 

The coronary artery calcification score, henceforth known as CACS, is calculated using cardiac computed-tomography (CT)-scan images with a limited radiation field (<1 mSv), taken without the injection of a contrast medium, and acquired during a 3–5 s breath hold [5]. The presence of calcification is quantified throughout the epicardial coronary artery tree. Coronary artery calcification is defined as a hyperintense lesion threshold > 130 Hounsfield units for an area of three adjacent pixels (at least 1 mm^2^). The original CACS, developed by Agatston et al. [6], is determined by multiplying the calcified plaque area by the maximum lesion density in each coronary artery.

The main objective of this work was to evaluate the CACS’s usefulness in the cardiovascular work-up of RA patients that are considered to be at low or intermediate cardiovascular risk. A CACS > 100 reflects notable atherosclerosis and requires complementary cardiological explorations.

## 2. Materials and Methods

### 2.1. Patient Selection

This descriptive, cross-sectional, monocentric, routine-care study was conducted from April 2020 to August 2021. All patients included in the study were informed about the study and its objectives, orally and in writing, with a letter of information and consent. They all gave their written consent for the anonymous use of their data. This study was conducted in accordance with the Declaration of Helsinki and Good Clinical Practices. The protocol was approved by the Amiens University Hospital Ethics Committee, i.e., the Department of Clinical Research and Innovation. A declaration of conformity to a reference methodology was made to the French National Commission on Informatics and Freedoms (CNIL) (Project identification code: PI2021_843_0090). 

Inclusion criteria were: age ≥ 18 years old; RA defined by the American College of Rheumatology/European Alliance of Associations for Rheumatology (ACR/EULAR) 2010 criteria [7]; in remission (Disease Activity Score (DAS)28 ≤ 2.6) or in Low Disease Activity (LDA, DAS28 ≤ 3.2) without any flare within the last 6 months; low or intermediate cardiovascular risk according to the Systematic Coronary Risk Evaluation (SCORE) [8]; affiliated with the French National Health Insurance. 

Non-inclusion criteria were: known diabetes, familial hypercholesterolemia, moderate-to-severe chronic renal insufficiency (clearance < 60 mL/min), malignant hypertension (BP ≥ 180/110), documented cardiovascular disease (coronary artery disease, peripheral arterial disease, stroke, transient ischemic attack), and/or opposition to the use of personal data for research purposes. 

### 2.2. Study Design

Patients satisfying the inclusion criteria and without conditions for non-inclusion were asked to come to an inpatient clinic for a full cardiovascular assessment. The work-up comprised: a collection of classic cardiovascular risk factors (CVRFs); RA characteristics and treatments; BP measurement (with a second verification if this was abnormal); weight and height measurements; laboratory work-up including a lipid profile, fasting blood sugar and inflammatory markers; electrocardiogram (ECG); an echo-Doppler test of the supra-aortic trunks (SATs); and a cardiac CT scan to determine the CACS. 

Patients with CACS > 100 were referred to the study’s cardiologist investigator, who completed the work-up with trans-thoracic echocardiography and stress echocardiography and determined if there was an indication for angioplasty. The rheumatologist managed other abnormalities and treatable CVRFs after consulting with multidisciplinary colleague(s). The patient’s primary-care physician was informed by letter to assure continuity of care. 

### 2.3. Data Recorded

Data were collected during routine clinical consultations or from electronic case reports. For each patient, the following information was recorded from their medical chart: demographic data: sex, age; classic CVRFs: sex and age (female > 60 years, male > 50 years), personal smoking history (active, former or non-smoker (never or stopped for > 3 years) and number of pack-years), dyslipidemia and treatment (number, medications), hypertension and treatment (number, medications), sedentary status according to occupation (active, sedentary defined as mostly sitting, unemployed/invalidity), family history of cardiovascular disease (MI or sudden death before 55 years of age of a 1st-degree male family member, MI or sudden death before 65 years of age of a 1st-degree female family member, early stroke before 45 years of age in 1st-degree relative(s)); RA characteristics: duration of disease (in months), immunonephelometry-detected rheumatoid factors (RFS; retained threshold > 15 IU/mL) and/or fluorometrically identified anti-cyclic citrullinated peptide-2 (CCP2) antibodies (retained threshold > 7 IU/mL) at diagnosis if possible, otherwise during the course of its progression, presence of erosions on the most recent radiographic images, all treatments received before taking biological Disease Modifying Anti-Rheumatic Drugs (bDMARDs) (current intake, number and types) and conventional synthetic Disease Modifying Anti-Rheumatic Drugs (csDMARDs): methotrexate, leflunomide, sulfasalazine, hydroxychloroquine, non-steroidal anti-inflammatory drugs (NSAIDs; current intake and cumulative doses), corticosteroids (current intake, prednisone equivalent, cumulative dose); weight and height (using the same measuring instruments) and body mass index (BMI) calculation (≥25 kg/m^2^: overweight; ≥30 kg/m^2^: obesity), systolic and diastolic BP; laboratory values: C-reactive protein (CRP; mg/L), erythrocyte sedimentation rate (ESR; mm at 1st hour), blood glucose (mmol/L), lipid analyses (total cholesterol, low-density lipoprotein (LDL), high-density lipoprotein (HDL) and triglycerides in g/L); calculated factors: DAS28 ESR and CRP [9], SCORE multiplied by 1.5, as recommended (recalculated according to the data obtained at the time of the cardiovascular work-up) [4]; complementary examination findings: ECG (normal, MI sequelae, cardiac hypertrophy, rhythm or conduction disorder), echo-Doppler test of the SATs (normal, arteriosclerotic plaques or significant stenosis), cardiac CT scan (CACS). The radiologist, blinded to the patients’ characteristics, designated each plaque on the scan images (GE LightSpeed VTC 64 TM with cardiac synchronization) and the CACS was calculated using the SmartScore application of AW VolumeShare 7 software (GE Healthcare, Chicago, IL, USA) that took into account plaque volume and density. The sum of the scores of all the plaques constituted the CACS expressed as an integer, as developed by Agatston et al. [6]. A second reading by a second radiologist, who was also blinded, was requested for high CACSs. 

### 2.4. Statistical Analyses

Quantitative variables were reported as mean ± standard deviation, median [Q1–Q3], and range. Categorical variables were reported as a number (percentage, calculated using the number of values available). The percentages of interest were given with a 2-sided 95% confidence interval (CI; Wilson’s method). Comparisons of quantitative variables between the different subpopulations of interest used the Wilcoxon tests; those for qualitative variables used the Chi-square or Fisher’s exact tests. A *p* < 0.05 was considered significant. Analyses were computed using SAS software version 9.4 (SAS Institute Inc., Cary, NC, USA).

## 3. Results

### 3.1. Patient Characteristics

The recruitment process identified 251 patients through coding (International Classification of Diseases 10th Revision, immunopositive RA and immunonegative RA) and 28 through routine consultations between April 2020 and June 2021. Two hundred and twenty-nine were not included for various reasons that are shown in Figure 1. Notably, with more advanced age came a systematic increase in their cardiovascular risk according to the SCORE chart, in accordance with the inclusion criteria, these so-called “advanced age” patients have been not included. Fifty RA patients were retained for this study and their characteristics are reported in Table 1.

### 3.2. Cardiovascular Assessment Results 

All 50 enrolled patients underwent a full cardiovascular work-up; their data are presented in Table 2.

### 3.3. CACS

Twelve (24% [95% CI: 14.3; 37.4]) patients had CACSs > 100, indicative of moderate arteriosclerosis burden and 7 (14% [7.0; 26.2]) patients’ CACSs exceeded 400, i.e., indicative of high plaque burden. The distribution of CACS values is shown in Table 3. The median non-zero CACS was 71 (range: 1 to 2008).

The 12 patients with CACSs >100 included four men (mean age: 60.3 ± 1.3 years) and eight women (mean age: 60.6 ± 3.7 years old). All but one patient were considered to have intermediate cardiovascular risk (mean SCORE: 3.3 ± 1.3 vs. SCORE: 1.4 ± 1.6 for the whole population). Already known CVRFs including hypertension were present in three patients, and dyslipidemia was present in five patients.

A CACS >100 led to the completion of complementary tests for myocardial ischemia. In one (8.3%) [1.5; 35.4] patient, trans-thoracic echocardiography detected a moderate dilatation of the ascending aorta, and for another the results of ECG-coupled stress echocardiography was positive, both electrically and sonographically. The stress echocardiography also revealed ventricular hyperexcitability which worsened during periods of peak exercise. BP increases during exercise were reported in three (25% [8.9; 53.2]) patients. The cardiologist ordered coronary angiography for two patients who underwent a stent insertion.

### 3.4. Other Cardiovascular Tests 

Forty (80% [67.0; 88.8]) patients had lipid profiles that showed elevated LDL and/or triglycerides. Six patients required immediate treatment because they had high LDL. Only 10 patients had lipid profiles within the target ranges. Two patients had fasting blood glucose levels ≥ 1.26 g/L. Upon repeat testing 3 months later, these values returned to normal for one patient and were again elevated for the other. The latter, who was diagnosed with diabetes, was asymptomatic with HBA1c > 10%. 

Eleven patients (22% [12.8; 35.2]) were hypertensive (systolic BP ≥ 140 mm Hg and/or diastolic BP ≥ 90 mm Hg). No severe hypertension was found. 

No MI sequelae were found. The four anomalies identified were simple conduction disorders that did not require specialist management or treatment. 

Atherosclerotic plaques on SATs were observed in 17 patients or 34.7% of the study population [22.9; 48.7]. No hemodynamically significant stenosis was found.

### 3.5. Atherosclerosis Assessment 

Atherosclerosis is classically assessed using a lipid profile and a SATs echo-Doppler test. As Figure 2 shows, only one patient in the CACS > 100 group had a lipid profile within the target ranges, meaning that 91.6% [64.6; 98.5] vs. 76.3% [60.8; 87. 0] in the ≤100 group had abnormalities; respectively, seven (58.3% [32.0; 80.7]) vs. 26.3% [15.0; 42.0] patients showed atherosclerotic plaques on the SATs echo-Doppler test. The CACSs led to the reclassification of five patients without atheromatous plaques in their SATs into the high cardiovascular risk group. Indeed, CACSs identified high-risk patients in a similar way to that of the echo-Doppler detection of atheromatous plaques on carotid or femoral arteries [8].

### 3.6. Factors Associated with Atherosclerosis 

As shown in Table 4, the CVRFs significantly associated with a high CACS were age according to sex and smoking status. No RA characteristics or treatments received were associated with the CACS, but a trend was observed for erosions (*p* = 0.067).

When analyzing those same factors to identify a potential relationship with lipid profile abnormalities or atheromatous plaques on the SATs echo-Doppler test, they were found to be significantly associated with BMI (*p* = 0.037, Fisher’s exact test) and erosions (*p* = 0.009, Chi-square test), respectively. 

## 4. Discussion

Herein, we evaluated the CACS contribution to the cardiovascular assessment of 50 RA patients at presumed low or intermediate cardiovascular risk. CACSs exceeded 100 for 24% and 400 for 14% of the patients, indicating moderate and high arteriosclerosis burdens, respectively. They led to the reclassification of five patients as being at high cardiovascular risk. Complementary examinations identified coronary ischemia that required angioplasty in two. CACS use seems relevant, especially for patients with intermediate cardiovascular risk, whose mean SCORE was 3.25 ± 1.3, compared to 1.4 ± 1.6 for the whole population.

Non-inclusion criteria for this study were carefully chosen. The absence of diabetes and moderate-to-severe chronic renal insufficiency allowed use of the SCORE chart [8]. Patients with documented cardiovascular disease were not included, as they fell into the very high cardiovascular risk group, for whom CACS is not recommended [10]. Also patients with stable RA or those in remission were selected to enable the reliable evaluation of lipid status [4].

The frequency of elevated CACSs in our patients is similar to those previously reported. Chung et al. conducted a comparative study on 227 patients who were divided into healthy, and newly diagnosed or established RA [11]. The frequency of CACS ≥ 110 in patients with inflammatory rheumatism was 29%, with a predominance of patients with a disease of >10 years’ duration. Most other studies did not use 100 as the cut-off to classify patients. For example, our team previously assessed coronary and aortic calcifications in 75 RA patients compared to 75 healthy controls [12], and found respective non-zero CACSs of 65.3% vs. 49.3%, respectively. However, that population was older (mean age: 60.7 years) than the one herein. A similar study by Wang et al. on groups of 85 patients each with a lower mean age of 53.9 years identified 41.2% with non-zero calcium scores, which is similar to our results [13]. 

To our knowledge, clinical consequences of a high CACS in such a population has not been previously examined. The majority of studies have used the CACS as a tool to reclassify patients’ cardiovascular risk [14]. Notably, two of our 12 patients with CACS > 100 had significant coronary ischemia that required angioplasty. In addition to assessing cardiovascular risk, CACS calculation also makes it possible to select more precisely those patients who need to be investigated with stress tests for silent myocardial ischemia [2,15]. We chose to use trans-thoracic echocardiography combined with ECG-coupled stress echocardiography because these imaging examinations are non-invasive and enable the researcher to visualize cardiac pathologies other than coronary artery disease, as seen in autoimmune diseases [16]. 

The evaluation of RA patients for atherosclerosis classically includes obtaining the lipid profile and employing echo-Doppler echocardiography of the SATs. In our study, abnormal findings for those two examinations were more often found in the high CACS group (≥100). CACS enabled independent reclassification of five patients without atherosclerotic plaques to the high cardiovascular risk group. Thus, it seems to provide a more accurate assessment that is focused on the coronary arteries. Observations reported in the literature have been mixed. In their analysis of 104 RA patients, Corrales et al. reported that a carotid ultrasound was more sensitive than the CACS at detecting subclinical atherosclerosis. However, patients at high and very high cardiovascular risk were included, and cardiac CT scans are not recommended for such individuals [17]. Most researchers found an ultrasound assessment of carotid or femoral plaque or CACS to be of comparable usefulness to predict cardiovascular events [8].

Concerning the other tests, 80% of our RA patients had lipid abnormalities. This high frequency could be explained by the fact that 17 patients treated with anti-IL6 and anti-JAK had been included and those treatments are known to alter the lipid balance. We also applied the most recent European Society of Cardiology recommendations, which used stricter values than previously included [8,18,19]. Inadequate lipid-abnormality management had previously been identified by the analysis of the application of the recommendations to the French COMEDRA (COMorbidities EDucation Rheumatoid Arthritis) cohort [20]. The rest of the laboratory work-up also enabled the detection of diabetes. Because insulin resistance is a known complication of RA [21], fasting blood glucose determination seems to be a relevant part of the work-up. ECG did not reveal any significant abnormalities in our study; hence, its usefulness seems moderate. In addition, 11 patients were hypertensive. Of course, a white-coat effect might explain this high number. Unlike dyslipidemia, hypertension is difficult to manage by a rheumatologist in daily practice. Finally, 17/49 (34.7%) patients had arteriosclerosis plaques on the SAT echo-Doppler tests. No significant stenosis was found. The frequencies reported in the literature varied widely [13,17]. We only assessed the presence or absence of plaque. Indeed, the measurement of carotid artery intima–media thickness was less informative than plaque detection [8]. 

In our study, age according to sex and smoking status were significantly associated with a high CACS. The former factors are well-known, with adjustment algorithms proposed for age, sex, and ethnicity [22]. The second factor is simply justified by the preponderant role of smoking in cardiovascular disease. Although no association between RA characteristics and higher CACSs was found herein, it is known that the positivity of any relevant antibody (ACPA or RF) and extraarticular involvement are associated with increased cardiovascular mortality [23,24]. In particular, ACPA-positivity is associated with impaired endothelial function, independent of other CVRFs [25]. Erosions were also significantly associated with atherosclerosis, which is easily explained by the link with RA activity [12]. Unsurprisingly, high BMI was significantly associated with more lipid abnormalities, which only reinforces our other findings [26].

Our study has several strengths, including thorough cardiovascular work-ups on a dedicated day and multidisciplinary consensus with the study cardiologist, vascular physicians, and radiologists. RA is not only an autoimmune rheumatism but also a systemic disease. Its daily management should therefore be as multidisciplinary as possible. Having the expertise of a cardiologist in the assessment and management of cardiovascular risk is a definite advantage, as our study shows. This study is in line with the latest European recommendations for cardiovascular risk management in rheumatic diseases [27]. Our study offers a new possibility for the early management of patients and a more relevant assessment of their risk. The centralization of the results and sending a letter to each patient’s primary-care physician assured the continuity of care. A limitation of our study is a lack of statistical power due to the small number of patients. A future multicenter study should seek larger recruitment.

## 5. Conclusions

The CACS obtained using cardiac CT-scan images is a non-invasive, rapid, and inexpensive examination. It enabled more accurate classification of cardiovascular risk and selection of patients eligible for screening for silent myocardial ischemia. It seems to provide additional information on coronary damage to initiate primary prevention, compared to the more commonly used lipid profile and the echo-Doppler test of the SATs to evaluate subclinical atherosclerosis. Its systematic use seems justified in the cardiovascular assessment of RA patients, particularly in those at intermediate risk according to the SCORE.

## Figures and Tables

**Figure 1 jcm-11-04841-f001:**
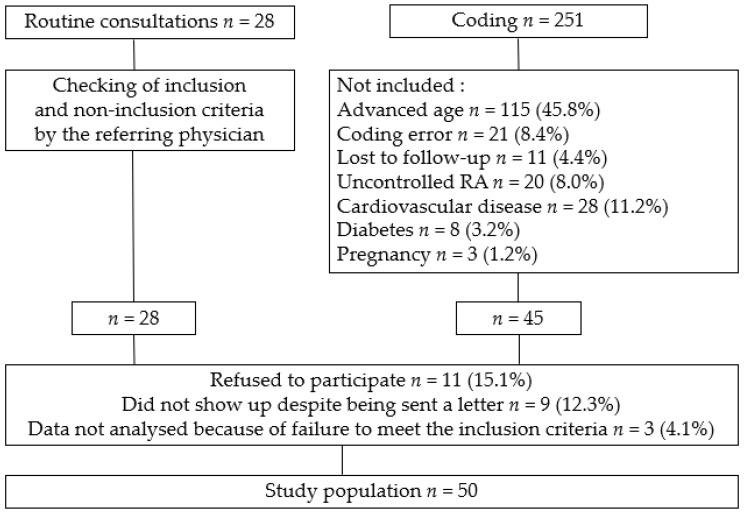
Flowchart.

**Figure 2 jcm-11-04841-f002:**
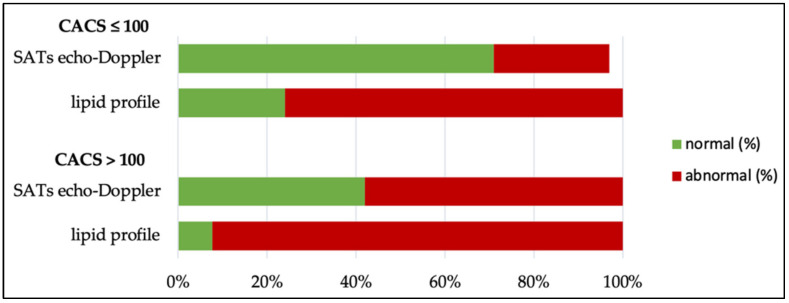
Lipid profile and SATs echo-Doppler abnormalities according to CACS.

**Table 1 jcm-11-04841-t001:** Characteristics of the 50 rheumatoid arthritis patients at the start of the study (baseline).

Characteristic	Whole Population	Missing Data
Female sex, *n* (%)	41 (82)	0
Age (years), mean ± SD	53.7 ± 7.5	0
Disease duration (months), median [Q1–Q3]	151.5 [70.0–238.0]	0
Rheumatoid factor-positive, *n* (%)	35 (74.5)	3
ACPA-positive, *n* (%)	34 (73.9)	4
Erosion presence, *n* (%)	26 (52)	0
**Ongoing** bDMARD, *n* (%)		0
None	10 (20)	
Anti-TNF	11 (22)	
Anti-IL6	17 (34)	
CTLA4-Ig	1 (2)	
Anti-CD20	5 (10)	
Anti-JAK	6 (12)	
Treatment duration (months), median [Q1–Q3]	38.5 [20.0–90.5]	
**Previously received bDMARDs**, *n* (%)		0
Anti-TNF	33 (66)	
Anti-IL6R	27 (54)	
Anti-CTLA4	12 (24)	
Anti-CD20	9 (18)	
JAK inhibitor	7 (14)	
Number of bDMARDs received, mean ± SD	2.2 *±* 2.0	
**Other ongoing treatments**, *n* (%)		
Methotrexate	36 (72)	0
Leflunomide	2 (4)	0
Sulfasalazine at present	0 (0)	0
Hydroxychloroquine ^a^	1 (2)	0
NSAIDs	4 (8)	0
Treatment duration (months), mean ± SD	54.5 *±* 7.8	2
**Corticosteroids**, *n* (%)	5 (10)	0
Dose (mg/d prednisone equivalent), mean ± SD	6.4 *±* 3.5	0
Treatment duration (months), median [Q1–Q3]	192.0 [38.0–198.0]	0

ACPA: anti-citrullinated peptides antibodies; bDMARDs: biological Disease Modifying Anti-Rheumatic Drugs; CD20: cluster of differentiation 20; CTLA4: cytotoxic T-lymphocyte antigen-4; IL6: interleukin-6; IL6R: IL6 receptor; JAK: Janus kinase; NSAIDs: non-steroidal anti-inflammatory drugs; Q1–Q3: first–third quartile; SD: standard deviation; TNF: tumor necrosis factor. ^a^ For another associated autoimmune disease.

**Table 2 jcm-11-04841-t002:** Cardiovascular assessment results for the 50 rheumatoid arthritis patients.

Cardiovascular Work-Up	Whole Population
**Risk factors**, *n* (%)	
MI or sudden death < 55 years of a 1st-degree male relative	5 (10)
MI or sudden death < 65 years of a 1st-degree female relative	1 (2)
Early stroke < 45 years in 1st-degree relative(s)	1 (2)
**Smoking**, *n* (%)	
Never or Stopped ≥ 3 years	11 (22)
Active	14 (28)
Number of pack-years (*n* = 25), median [Q1–Q3]	20.0 [10.0–40.0]
**Sedentariness**, *n* (%)	
Sedentary work	20 (40)
Unemployed/invalidity	13 (26)
Treated dyslipidemia, *n* (%)	13 (26)
Treated hypertension, *n* (%)	10 (20)
**Clinical parameters**, median [Q1–Q3]	
Weight (kg)	75.0 [63.0–90.0]
Height (cm)	167.0 [162.0–174.0]
BMI (kg/m²), median [Q1–Q3] *m ±* SD	27.9 ± 7.5
Systolic blood pressure (mm Hg)	131.0 [120.0–138.0]
Diastolic blood pressure (mm Hg)	76.0 [70.0–83.0]
**Laboratory tests**, mean ± SD	
Total cholesterol (g/L)	2.1 *±* 0.4
LDL (g/L)	1.4 *±* 0.4
HDL (g/L)	0.6 *±* 0.2
Triglycerides (g/L)	1.3 *±* 0.7
Fasting blood glucose (mmol/L)	5.1 *±* 1.9
**Complementary investigations**, *n* (%)	
ECG abnormality	4 (8)
Atherosclerosis on SATs ultrasound	17 (34.7) ^a^
CACS > 100	12 (24)
**RA activity**, mean ± SD	
DAS28 CRP	2.1 *±* 0.7
DAS28 ESR	2.3 *±* 0.9
**SCORE****×****1.5**, mean ± SD	1.4 *±* 1.6
0, *n* (%)	24 (48)
1.5, *n* (%)	10 (20)
3, *n* (%)	10 (20)
4.5, *n* (%)	6 (12)

CRP: C-reactive protein; DAS: disease activity score; ESR: erythrocyte sedimentation rate; MI: myocardial infarction; BMI: body mass index; LDL: low-density lipoprotein; HDL: high-density lipoprotein; ECG: electrocardiogram; Q1–Q3: first–third quartile; SATs: supra-aortic trunks; SD: standard deviation. ^a^ Not achieved for one patient

**Table 3 jcm-11-04841-t003:** CACS values for the 50 rheumatoid arthritis patients.

CACS	*n* (%)	95% CI
0	25 (50)	[36.6; 63.4]
[1–100]	13 (26)	[15.9; 39.6]
[100–400]	5 (10)	[4.3; 21.4]
>400	7 (14)	[7.0; 26.2]

**Table 4 jcm-11-04841-t004:** Factors associated with a CACS > 100.

Associated Factors	CACS ≤ 100 (*n* = 38)	CACS > 100 (*n* = 12)	*p*
**Cardiovascular risk factors**, *n* (%)			
≥1 family member with cardiovascular disease	4 (10.5)	2 (16.7)	0.621 (F)
Female > 60 years or male > 50 years	6 (15.8)	10 (83.3)	**<0.001** (F)
Smoking ^a^	16 (42.1)	9 (75)	**0.047** (C)
Sedentariness ^b^, *n* (%)			0.237 (F)
Active	15 (39.5)	2 (16.7)	
Sedentary work	15 (39.5)	5 (41.7)	
Unemployed/invalid	8 (21.1)	5 (41.7)	
BMI (kg/m^2^), *n* (%)			0.917 (F)
Normal < 25	13 (34.2)	5 (41.7)	
Overweight [25–30]	13 (34.2)	3 (25)	
Obese ≥ 30	12 (31.6)	4 (33.3)	
Treated dyslipidemia, *n* (%)	8 (21.1)	5 (41.7)	0.256 (F)
Treated hypertension, *n* (%)	7 (18.4)	3 (25)	0.686 (F)
**RA characteristics**			
Disease duration (months), median [Q1–Q3]	151.5 [70.0–238.0]	163.5 [77.5–244.5]	0.683 (W)
Rheumatoid factor-positive ^c^, *n* (%)	28 (77.8)	7 (63.6)	0.435 (F)
ACPA-positive ^d^ *n* (%)	25 (71.4)	9 (81.8)	0.701 (F)
Erosion(s) present, *n* (%)	17 (44.7)	9 (75)	0.067 (C)
DAS28 CRP, median [Q1–Q3]	2.0 [1.5–2.6]	2.2 [1.5–2.7]	0.725 (W)
DAS28 ESR, median [Q1–Q3]	2.2 [1.5–2.9]	2.2 [1.6–3.1]	0.991 (W)
**Ongoing Treatments**, *n* (%)			
bDMARDs	30 (78.9)	10 (83.3)	1.000 (F)
Methotrexate	25 (65.8)	11 (91.7)	0.140 (F)
NSDAIs	3 (7.9)	1 (8.3)	1.000 (F)
Corticosteroids	3 (7.9)	2 (16.7)	0.582 (F)

95% CI: 95% confidence interval; ACPA: anti-citrullinated peptide antibodies; bDMARDs: biological Disease Modifying Anti-Rheumatic Drugs; BMI: body mass index; (C): Chi-2 test; DAS: Disease Activity Score; (F): Fisher’s exact test; NSAIDs: non-steroidal anti-inflammatory drugs; RA: rheumatoid arthritis; (W): Wilcoxon test. ^a^ Active smoker or stopped smoking. ^b^ Sedentary lifestyle according to employment. ^c^ Missing data for 4 patients. ^d^ Missing data for 3 patients.

## Data Availability

Not applicable.

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
