# Peer review of "Is the Calcium Score Useful for Rheumatoid Arthritis Patients at Low or Intermediate Cardiovascular Risk?"

_jcm, 2022, doi:10.3390/jcm11164841_

Round 1
Reviewer 1 Report
The study show data about a careful and extensive evaluation of cardiovascular risk in patients with “stable” RA or in remission. The design is a small case series. The aim of the study was to do a complete work-up included collection of cardiovascular risk factors, laboratory 23 analyses, electrocardiogram, supraaortic trunks (SATs) echo-Doppler and cardiac CT scan with determination of de CACS. In addition, it shows that the use of coronary artery calcification score calculated on CT-scan images may be useful in the evaluation of CV risk in this type of patients. However the diagnostic utility of the CACS was not evaluated, because this type of approach requires another type of experimental design i.e a Diagnostic test utility approach.
They found a high CACS (>100) it means a moderate 24% or high 14% burden of atherosclerosis in 11 patients with intermediate risk according to the SCORE risk classification. And they reclassified as high risk in 5 of those patients.
The sample of patients included is of 50 middle aged women, with 12 to 20 years of RA, 75% ACPA or RF positive, 80% with bDMARDs or sdDMARDS and 78% in combo with csDMARDs and 10% with prednisone.
Inclusion criteria:
Is not clear for me what “stable or in remission RA based on Disease Activity Score-28 (DAS28 calculation for at least 6 months” means. For a Rheumatologist remission according DAS28 calculated using erythrocyte sedimentation rate and four variables (SJC, TJC, Global health and ESR) means a DAS ≤2,6. But the “stable” term should be clarified. I suppose it means that remission is maintained in the previous 6 months. Please clarify it.
The authors identified 251 patients and for various reasons thy end up analyzing 50 patients. They should put the n and % of patients who leak if is possible on a flow chart. This filter is important for external validity of the data. Do you know the demographic characteristics of the 201 patients not ncluded?? They were similar to the included patients??
The relevance of this study beyond the calculation of the CACS, is the added value of incorporate a cardiologist in the team. And of course the management of the comorbidities in the long standing RA patients with strict outcome measures (Lipids, hypertension, smoking, diabetes, insulin resistance, obesity, sedentarism, etc) by a multidisciplinary team.Author Response
Please see the attachment.

Reviewer 2 Report
Introduction:
Please mention how cardiovascular risk is usually assessed in rheumatoid arthritis.
Materials and methods:
“duration of disease progression (in months),”
Comment: Do you mean “disease duration”?
“all treatments received before biotherapy (current intake, number and types)”
Comment: Do you mean biological therapy?
“corticosteroids (current intake, specialty, dose, cumulative dose)”
Comment: What is meant by “corticosteroid specialty”?
Discussion:
“CACS use seems relevant, especially for patients with intermediate cardiovascular risk,
whose mean SCORE was 3.25 ± 1.3 compared to 1.4 ± 1.6 for the whole population.”
Comment: This statement has not been mentioned in the results.
“No use other classification criteria were used for this parameter. Indeed, the measurement of carotid artery intima–media thickness was less informative than plaque detection [8].”
Comment: The first sentence is not clear.
“Erosions were also significantly associated with atherosclerosis, which is easily explained by the link with RA activity. Unsurprisingly, high BMI was significantly associated with more lipid abnormalities, which only reinforces our other findings”
Comment: Please cite the relevant references.
References:
Reference [8]: The authors are missing.
Mach F, Baigent C, Catapano AL, Koskinas KC, Casula M, Badimon L, et al. 2019 ESC/EAS Guidelines for the management of dyslipidaemias: lipid modification to reduce cardiovascular risk. Atherosclerosis. 2019; 290:140-205.
Reviewer 3 Report
no commnets
Author Response
We thank reviewer 3 for his very good evaluation of our study.